# Development of Pilgrimage Tourism in Slovakia over the Past Decades: Examples of Selected Pilgrimage Sites

**Miloš Jesenský** [1], **Enikő Kornecká** [2], **Mário Molokáč** [2,*] and **Dana Tometzová** [2]

1. Belianum-Matej Bel University Press, Matej Bel University, Tajovského 51, 97401 Banská Bystrica, Slovakia; milos.jesensky@umb.sk
2. Department of Geo and Mining Tourism, Institute of Earth Resources, Faculty of Mining, Ecology, Process Control and Geotechnologies, Technical University of Košice, 04200 Košice, Slovakia; eniko.kornecka@tuke.sk (E.K.); dana.tometzova@tuke.sk (D.T.)
* Correspondence: mario.molokac@tuke.sk

**Abstract:** Pilgrimage tourism, among the earliest forms of tourism with a tradition spanning centuries, stands as a steadfast global attraction. This article delves into the significance, diversity, and historical roots of this tourism type, recognizing the contemporary surge in interest in pilgrimage sites. Offering an overview of globally prominent pilgrimage destinations and highlighting key locations in Slovakia, the article emphasizes the country's considerable potential for the utilization and development of these sacred sites, despite its compact size. It stresses the need to foster this historically significant tourism sector and the necessity for attention and support from the government sector to maximize its potential. The relevance of pilgrimage became particularly apparent during the COVID-19 pandemic, with observable visitor participation despite challenging conditions at various pilgrimage sites. The article examines the pilgrimage's evolution before, during, and after the pandemic, using Levoča and Šaštín in Slovakia as illustrative cases. One of the main objectives of this study was to clarify the development of pilgrimage tourism in Slovakia over the past decades and the factors influencing it. The attendance analysis unmistakably reveals a significant upward trend at these specific locations, emphasizing the need to establish collaborative efforts to support this sector. Such collaboration is crucial for ensuring the sustainability of historically significant sites, fostering local development, and increasing the visibility of less-visited pilgrimage destinations.

**Keywords:** pilgrimage tourism; pilgrimage sites; COVID-19 pandemic; attendance; pilgrimages

## 1. Introduction

Travel and tourism represent highly dynamic segments of the economy, both in Slovakia and globally. The tourism industry plays a significant role in the economy, primarily as a source of income. It caters to the needs of travelers and aims to make the most efficient use of the area's natural, cultural–historical, and human potential.

Religious tourism, also known as pilgrimage, sacred, faith, or spiritual tourism, is a widespread form of tourism [1]. Pilgrimage was the first type of tourism mobility to come into existence thousands of years ago [2]. Pilgrimage is a cult practice of most world religions and, therefore, a phenomenon that has been studied since time immemorial. Pilgrimage sites are a place of "revelation". Thanks to the number of visitors, they have been transformed into pilgrimage centers. Pilgrimages, as a specific, religiously motivated type of migration, are the subject of pilgrimage research [3].

According to Gladstone [4], pilgrimage tourism involves traveling to temples, shrines, or other sites of significance to the faith and beliefs of travelers. This concept is often associated with religious tourism.

Starting from the 2000s, scholars have expanded the definition of pilgrimage to encompass not only traditional religious expeditions but also contemporary secular travel. This shift reflects a broader discussion among researchers, who now consider modern

pilgrimage to be within the framework of spiritual motivations rather than solely religious ones. As evidenced by numerous studies, an increasing number of travelers are pursuing diverse experiences, such as enlightenment, learning, enhanced spiritual and physical health, and adventure. During this period, scholars have produced new insights into secular pilgrimage destinations and non-religious dimensions of pilgrimage studies [5]. Contemporary literature views pilgrimage as a comprehensive concept with both religious and secular roots, encompassing sites originating from diverse religious and non-religious backgrounds. Pilgrimage is undergoing a phase of renewal, resulting in the diminishing of certain unique characteristics, particularly its religious elements, which traditionally delineated it as a distinct type of tourism. Simultaneously, it is evolving new identities, including secular pilgrimage, spiritual, religious, church, dark, and transformational tourism [6].

Religion or faith is perceived as a belief system rooted in tradition, encompassing beliefs about the universe and the powers governing it. Within this framework, it signifies humanity's connection to the concept of the sacred, which is manifested through religious doctrine, worldview, and structure [7]. Matlovičová et al. [8] defined religious tourism as encompassing all tourist travels that is primarily motivated by visits to sacred and religious sites or towns. Unlike pilgrimage tourism, it constitutes a broader category. For instance, religious tourism may involve adherents of religious faiths that lack a tradition of pilgrimage within their religious practice, such as numerous Protestant denominations [8]. Pilgrimage tourism constitutes a subset of religious tourism, characterized by travel that is primarily motivated by religious or religious–cognitive purposes, with a significant portion of the journey dedicated to engaging in pilgrimage activities within a religious context. Pilgrimages are typically organized and adhere to stringent ritual protocols, incorporating various religious rituals such as prayers, worship, meditation, and other ceremonial practices. These journeys often lead to sacred destinations, including cult centers or sanctuaries [8]. In alignment with the commonly acknowledged geographical delineation of tourism, pilgrimage tourism can be defined as a subset (subtype) of religious tourism. Pilgrims, the participants of this form of tourism, temporarily depart from their habitual residence to visit a sacred (holy) site for religious or spiritual purposes. During their visit, they engage in activities associated with religious worship [3].

Participants in religious tourism travel with the purpose of visiting sacred and pilgrimage sites as well as religious landmarks. Often, they come to these places to commemorate significant religious events. The goal of visitors is to explore the culture, the nature of the world's religions, and their traditions. Religious sites, where pilgrimages and rituals are regularly performed, are considered the most common destinations. Religious tourism is often accompanied by various legends and stories about saints. Pilgrimage sites can also include rivers and mountains. Religious motifs dominate in pilgrimage tourism, which is the main difference between religious tourism and cultural exploratory tourism. Pilgrimages are a specific, religiously motivated form of migration that has accompanied humanity since its earliest history, with a significant impact on many countries [9]. Religious tourism is a form of travel exclusively or strongly motivated by religious reasons. It is one of the oldest forms of tourism and a global phenomenon in religious history, with various manifestations. Short-term religious tourism involves excursions to nearby pilgrimage centers or religious conferences. Long-term religious tourism entails visits to national and international pilgrimage sites or conferences lasting several days or weeks. Organizational forms of religious tourism can be distinguished based on definitive characteristics such as the number of participants, transportation choices, seasonal travel, and social structure [10].

According to Rinschede [10], religious tourism encompasses any form of tourism in which participants are exclusively motivated by religious reasons during their journeys, particularly during their stays at specific destinations. There are two categories of participants. The first category predominantly involves the arrival of the faithful at religious holiday destinations (such as visits to religious centers by pilgrims), which includes participation in ceremonies like church consecrations and spiritual exercises, among others. The second category consists of the arrival of a narrowly defined professional audience (clergy)

to religiously motivated, often exclusive, conferences. A similar perspective is presented by Vukonić [11], who delineates religious tourism as comprising three elements: pilgrimage (involving continuous individual or group visits to sanctuaries), "large-scale events" (held on significant religious anniversaries or occasions), and tours to significant religious sites and objects as part of a tourist itinerary, regardless of the timing of the journey.

Since the early days of Christianity, particularly after 313 AD, a dense network of pilgrimage routes has crisscrossed Europe. Along these routes, Christians have journeyed from the east to the west and from the north to the south. Numerous groups of pilgrims have visited places associated with the life and death of Jesus Christ [12]. Large groups of pilgrims have even reached Santiago de Compostela, the final resting place of the apostle St. James the Greater, situated at finis terrae—the westernmost point in Europe. The French city of Tours, where St. Martin died in 397, was also a highly frequented destination in the early Christian and early medieval periods. Pilgrims have also made their way to the tombs of St. Patrick, St. Boniface, and St. Olaf [13]. Christianity formed the population and territory of Central Europe, including Slovakia, from the 9th century. Its influence has been shown in so many places within the landscape to the degree that it has acquired the specific appearance of a religious landscape filled with religious buildings, objects, small sacral architecture, and Christian symbols accompanied by typically adjusted vegetation—gardens, trees, and solitary trees. In Slovakia, two dominant religious landscaping complexes have gradually become part of the area included in the UNESCO World Heritage List [14,15].

In recent years, the segmentation of religious tourism and the development of new market forms have led scholars to consider different types of religious tourists, such as spiritual tourists [16], "New Age" spiritual travelers [17], and "cyber-pilgrims" [18]. For example, Digance [18] noted that "cyber-pilgrims" can perform certain rituals as part of their online experience, like making a cup of coffee, lighting a candle, or burning incense at a specific time. The earliest studies on religiously motivated travel emerged in various tourism journals in the 1980s [19–21]. Subsequently, a typology for religious tourism was developed based on the evolution of pilgrimages in various religious directions [22]. The connection between tourism and pilgrimage was introduced in the literature in 1969, characterizing pilgrimage as the oldest form of this type of tourism [23–25]. Travelers expressing religious motivations have a strong inclination to embark on pilgrimages or to visit religious sites [26–28]. The clear market potential has spurred interest in religious tourism, which encompasses those who "repeatedly visit religious sites and/or make pilgrimages" [29]. The most well-known classification of religious tourist attractions was published by Nolan and Nolan [22], who proposed a classification of religious tourism resources based on three overlapping types: pilgrimage shrines, religious tourist attractions, and locations where religious festivities take place. The distinction between pilgrimage shrines and religious tourist attractions lies in the fact that the former are the focus of pilgrimage routes and have limited tourism value, while the latter are visited by both tourists and believers but are not considered pilgrimage sites. Nolan and Nolan [22] include most monasteries and cathedrals in this second category, although there are also pilgrimage shrines that qualify as tourist attractions. In reality, it is not unusual for tourists to outnumber pilgrims at these sites because they are often known for their art, architecture, and other elements. Religious tourism takes on many more forms than just pilgrimages. For example, Cohen [30] explains that many people who visit Israel go there not only for the holy sites but also for the Jewish atmosphere. He also notes that even though visits to major Christian cathedrals are not pilgrimages, they should certainly be considered religious tourism [30]. Interestingly, the most-visited tourist attraction in Europe is Notre-Dame de Paris, with 13 million visitors per year, and six other churches—Mont Saint-Michel, Basilica of the Sacré-Cœur, and the cathedrals in Reims, Chartres, Vézelay, and Sainte-Chapelle—are among the top 20 most visited places in France [31]. As Aniqa Ijaz of the University of Okara in Pakistan points out, the process of observing and practicing religious activities during visits to any sacred place, combined with fervent faith, can be categorized as religious tourism. Many researchers have explained its various types from their perspectives. Within the framework of

modern scientific society, modern religious tourism can be divided into two main types. One is pilgrimage, and the other is intellectually educational excursions. The second major category has two branches. These intellectually educational excursions are based on two characteristics of tourists: scientific knowledge and visits to religious events [32].

Religious tourism is one of the oldest forms of tourism we know. One of the reasons for this is likely its close connection to history. We recognize this type of tourism as part of cultural and educational travel. Horodníková and Derco [33] also state that pilgrimage tourism constitutes a segment of cultural tourism, alongside other forms such as dark, arts, heritage, creative, craft, and gastronomy tourism. Religious travel helps deepen our knowledge in areas such as history, culture, and faith. For many people, religion is a way of life. It establishes the norms by which the faithful conduct themselves. Church landmarks significantly contribute to our understanding of historical interpretation and the past, thus leading to new insights that allow people to form a relationship with their history and cultural and religious heritage. Understanding history also helps foster harmony among local residents and their cultural heritage. Religious tourism serves as an effective tool for advancing culture, faith, and tourism, a concept delineated by three motivational dimensions. The first pertains to spiritual–religious motivation, the second to cultural–educational motivation, and the third to spiritual–relaxation motivation. Understanding and distinguishing these motivational dimensions contributes to a deeper comprehension of contemporary religious tourism. A significant aspect of the current development of this form of tourism is that individual motivations are not strictly delineated but can be interconnected. For example, a tourist exploring the culture and history of religion itself is motivated not only by a desire for education but may also seek the spiritual–relaxation aspect of this experience and may also have a religious inclination. Conversely, a tourist strictly oriented toward religion is motivated by a desire for a deeply spiritual–religious experience. The historical development of this form of tourism is not only closely related to the development of society, the individual's personality, and societal events. The diversity of motivations reflects the complexity of contemporary demands and also the tourists' expectations for their experience. In contrast to the past, contemporary religious tourism has undergone a transformation into a more modern and intricate form.

Religious travel has had relatively favorable conditions for its development, but it mainly depends on good organization, intense collaboration among organizers, and the creation of an attractive product. However, a limiting factor is often the lack of financial resources. According to Kapur [34], the main benefits of religious tourism include strengthening faith, preserving and appreciating heritage, creating new "attractions", and opportunities for economic and social development.

Religious travel, along with pilgrimage, is one of the oldest forms of travel, and besides its spiritual and educational aspects, it also makes a significant contribution to the economy. Food and accommodation services can participate in it. The main motive for this type of tourism is visiting sacred places, churches, monasteries, pilgrimage sites, religious landmarks, and religious events. Visitors aim to understand the culture, world religions, and their traditions. Pilgrimage sites, where pilgrimages and rituals are regularly held, are among the most common. Pilgrims who come for religious celebrations from greater distances, for example, book accommodation and stay longer. They are also interested in other beauties and attractions, such as natural or cultural values [34].

## 2. Materials and Methods

The analysis encompasses the territory of Slovakia, with its pilgrimage background, and it seeks to highlight the development of pilgrimage tourism and the factors influencing its evolution. In the first step, pilgrimage sites and their significance were analyzed, followed by a selective method to identify the most notable pilgrimage sites for further examination. Visitor attendance was selected as an indicator of their tourism development. In the subsequent step, efforts were made to obtain this parameter from the selected pilgrimage sites. The initial challenge arose from several pilgrimage sites not recording visits,

providing only rough estimates which were insufficient for our purposes. Consequently, considering their significance, we reduced the selection to two pilgrimage sites: Levoča and Šaštín. Detailed pilgrimage data were gathered from multiple sources, prioritizing their credibility. Initially, we reached out to authorities/organizations involved in organizing pilgrimages. Additional information was obtained from press releases in online media, supplemented by personal interviews with staff from individual pilgrimage sites or those directly involved in organizing pilgrimages. In this way, data were collected in sufficient quality. The gathered data were structured and systematized. Descriptive and analytical methods were employed to present the research results.

One of the main objectives of this study was to clarify the development of pilgrimage tourism in Slovakia over the past decades and the factors influencing it. It was hypothesized that people turn to God during difficult periods. The COVID-19 pandemic was such a period, which, however, was unfavorable for organizing mass events such as pilgrimages. Therefore, we anticipated that after the restrictions were lifted, visitation would increase compared to before the COVID-19 pandemic. Finally, we aimed to generalize this development and demonstrate its similarity, at least to Christian pilgrimage tourism in Europe.

## 3. Analysis of Pilgrimage in Slovakia

Given the topic at hand, it is crucial to begin by elucidating the fundamental concepts that will be central to this article. In accordance with the pertinent legislation currently in effect in Slovakia, cultural heritage is defined as a tangible or intangible asset officially designated as such for the purpose of protection. Furthermore, there exists a higher level of heritage protection, designated for those of national cultural significance, for specific objects that are termed national cultural heritage sites. Most national cultural heritage sites also possess the status of cultural heritage. However, they can differ in terms of demarcation. For example, a national cultural heritage site may only encompass the most valuable part of a cultural heritage complex. Conversely, a national cultural heritage site can include multiple individually registered cultural heritage sites. Significant characteristics of the cultural heritage sites under examination are their sacred nature, their intensive use for religious purposes, and their role in serving pilgrimage tourism.

In the context of this article, pilgrimage tourism is understood as a journey externally leading to a sacred place and internally leading to spiritual understanding, according to the definition provided by Collins-Kreiner et al. [35]. Alternatively, as articulated by Bhardwai and Rinschede [36], pilgrimage involves leaving one's place of permanent residence for the purpose of performing religious practices. Pilgrimage tourism also encompasses a theological dimension, as described by Gavenda [37], signifying an activity directed toward sacred sites for religious purposes. These purposes include witnessing, praying, venerating, fulfilling specific obligations, and seeking the fulfillment of supplications.

For the purposes of our study, we will interpret pilgrimage tourism in a broader sense, encompassing not only travel motivated exclusively by religious reasons but also the exploration of sacred sites that are simultaneously cultural heritage sites. Within this scope, our focus will be on historical cultural heritage sites related to pilgrimage tourism that were constructed before the end of the 19th century. The topic of religious or pilgrimage tourism has gained increasing attention from domestic researchers in recent years in the context of its development in Slovakia. There is no shortage of such studies; for instance, Krogmann [38] conducted an analysis of this phenomenon in a specific geographic region, namely the Nitra Self-Governing Region. Another author, Čuka [39], examined pilgrimage activities in the village of Staré Hory, with a focus on the size, intensity, seasonality, and geographical background of the pilgrimage tourism. Subsequent research by other scholars [40,41] (Bubelíny 2008, Čuka, Bubelíny, Gregorová 2009) enriched our knowledge of the utilization of a particular pilgrimage site in the present with valuable information. Similarly, pilgrimage tourism has been analyzed, for example, in the village of Rajecká Lesná, considering the structure and frequency of visits, as well as other significant indicators [41].

Levoča, which is internationally significant, along with Šaštín, Nitra, and Staré Hory, which are nationally significant, are recognized as the most prominent pilgrimage sites in Slovakia [42]. Consequently, they hold a central role in our study. At the outset, special attention was also given to the concept of routes connecting historical sacred objects and building upon historical pilgrimage routes.

### 3.1. Pilgrimages and Pilgrimage Routes in Slovakia

Similar to examples abroad (with the Camino de Santiago being a typical instance), the cultural and historical value of sacred sites is heightened by their integration into the landscape, all while maintaining their original character and interconnectedness. The religious and esthetic perception of these sites is strengthened by their interconnectedness within a defined route. An analogous project to the Camino de Santiago leading to the Cathedral of St. James in Santiago de Compostela is the unique project of the Way of St. James in Slovakia, summarized in Table 1. It connects historical pilgrimage sites and, in addition to religious motifs, allows for an introduction to significant sacred sites from an art and architectural history perspective. Its main branch begins in Košice, traverses Slovakia, passing through Bratislava, and extends into Austria and Hungary, where it connects to the European network of pilgrimage routes to Santiago de Compostela. On Slovak territory, it measures 621 km and can be completed in approximately 30 days.

**Table 1.** Description of the Way of St. James in Slovakia (elaborated by authors according to [43]).

| Route Leg | Phases of the Route | km | Characteristics of the Route Leg |
| --- | --- | --- | --- |
| Košice—Levoča | 5 | 100 | From Košice (St. Elisabeth Cathedral) through Gelnica (Church of the Assumption of the Virgin Mary), Krompachy (Church of St. John), Žehra (Church of the Holy Spirit), Spišská Kapitula (Cathedral of St. Martin), Mariánska hora (Basilica of the Visitation of the Blessed Virgin Mary) to Levoča (Basilica of St. James). Here, a branch of the route connects from the place of Marian apparitions on Zvir Mountain in Litmanová (Chapel of the Apparition) and Červený Kláštor. |
| Levoča—Donovaly | 9 | 173 | From Levoča to Kežmarok (Basilica of the Holy Cross), then through the High Tatras, Važec (Church of the Adoration of the Three Kings), down the Váh valley through Liptovský Mikuláš (Church of St. Nicholas) to Donovaly (Church of St. Anthony of Padua). |
| Donovaly—Hronský Beňadik | 6 | 141 | From Donovaly through Špania Dolina (Church of the Transfiguration of the Lord), Banská Bystrica (Church of the Assumption of the Virgin Mary), Zvolen (Church of St. Elizabeth), Banská Štiavnica (Church of Our Lady of Mount Carmel) to Hronský Beňadik (Monastery of St. Benedict). |
| Hronský Beňadik—Trnava | 6 | 126 | From Hronský Beňadik through Topolčianky (Church of St. Catherine of Alexandria), Nitra (Cathedral of St. Emeram), Sereď (Church of St. John the Baptist) to Trnava (Church of St. James). |
| Trnava—Bratislava | 4 | 83 | From Trnava through Červený Kameň (Church of the Immaculate Conception of the Virgin Mary), Pezinok (Church of St. Sigismund), Marianka (Basilica of the Nativity of the Virgin Mary) to Bratislava (St. Martin's Cathedral). |

From Bratislava, a short 11.5 km route section leads to Wolfsthal in Austria, where it connects to the Austrian branch of the Way of St. James (Jacobsweg). Similarly, from Bratislava, it is possible to connect to the Hungarian part of the Way of St. James (Szent Jakab út) via a 25 km section through Rusovce.

In Slovakia, the official route of the Way of St. James displayed in Figure 1 was established through the initiative of Gerhard Weag from the Forum of the Way of St. James,

Friends of the Way of St. James, and partners of Jakobswege Wien—Salzburg Austria. After an international meeting in Vienna and the approval of the I-20 route (Camino Europe), construction began in the spring of 2012 in the section from Perlová Dolina to Plejsy. During the European Capital of Culture Košice 2013, the Košice—Kojšovská hoľa—Thurzov—Plejsy—Spiš Castle—Levoča route was officially opened with the participation of Bishop Milan Lach, along with mayors of surrounding towns, on 17 November 2013. In the spring of 2014, the second route, Levoča—Kežmarok—Podolínec—Litmanová, was opened and blessed in Levoča. The connection of the Way of St. James to Austria (Wolfsthal—Wien section) took place in Bratislava at the Theological Faculty of Trnava University under Dean Milan Lichner on 20 May 2015. The Trnava (St. James' Church) —Svätý Jur—Bratislava route of the Way of St. James was officially opened in 2015. The ceremonial connection of the Way of St. James in Slovakia and Poland (Litmanová—Eliášovka—Starý Sącz) took place on 29 August 2015, with the participation of three bishops.

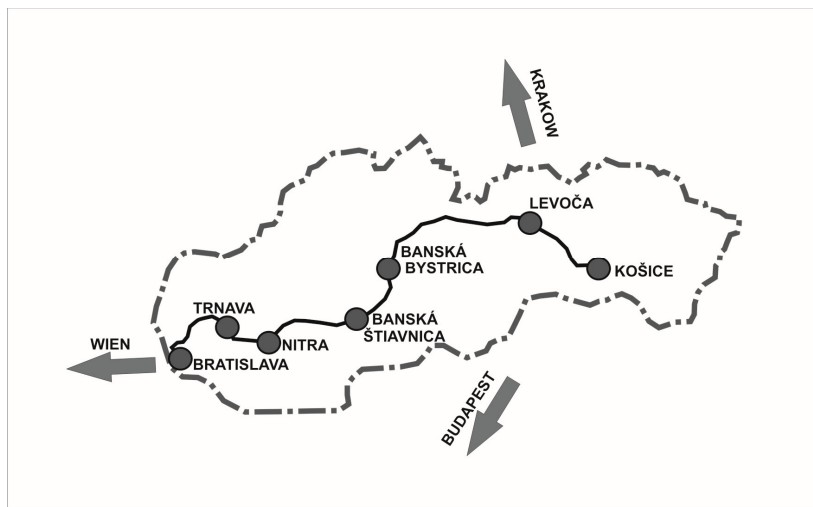

**Figure 1.** The Way of St. James in Slovakia (elaborated by authors according to [43]).

The Way of St. James has three separate routes in Slovakia. The eastern route begins in Košice on Komenského Street near the House of the Jesuits and continues through Čermeľ and Bankov to Jahodná, then through Predné Holisko and Kojšovská hoľa (1246 m above sea level) and continuing on to Perlová Dolina to Gelnica. From Gelnica, the Way of St. James leads through Thurzov and Plejsy to Spiš Castle and further to Levoča. The entire route is marked with a yellow arrow, and along the route, there are informational boards with a description of the location. The second part of this eastern route, referred to as the Spišská route, leads from Levoča to Kežmarok (where the route is marked with St. James' shells). It then proceeds through Podolínec, Vyšné Ružbachy Spa, and Litmanová to Eliášovka, where it intersects with the route of the Way of St. James in Poland (Droga Sw. Jakuba). Alternatively, you can travel along the Way of St. James from Červený Kláštor through Litmanová to Eliášovka (1023 m above sea level), or, starting from Litmanová, embark on a full-day journey through Lesnica to Červený Kláštor.

The northern route starts in Žilina near Budatín Castle and leads through Kysucké Nové Mesto (Church of St. James the Elder) and Živčáková Mountain near Turzovka, ultimately reaching Trojmedzie in the Kysuce region. This is the location where the borders of Slovakia, Poland, and the Czech Republic meet at a single point.

The western route starts from the historic center of Trnava at the Franciscan Church of St. James, passing through vineyards and forests in the Small Carpathians, passing through Pezinok (Capuchin Monastery), the town of Svätý Jur, and the Kamzík hill before reaching Bratislava. In Bratislava, archeological research revealed the historical foundations of the Church of St. James near the Old Market in the Old Town. From Bratislava's Petržalka district, the route leads through the village of Wolfsthal into other regions of Austria, where

it connects to the European pilgrimage route to Santiago de Compostella. In addition to the Way of St. James, Slovakia also features the Marian Route, which connects significant Marian pilgrimage sites. It begins at the parish church of St. Vojtech in Gaboltov, near Bardejov (49.366920° S, 21.142673° W). This church, with the first written mention dating back to 1247, is the most significant pilgrimage site in the Košice Archdiocese. As early as 1706, Eger Archbishop Stephen Telekessy (1633–1715) designated it as a pilgrimage site. According to canonical visitations in the second half of the 18th century, pilgrimages were an integral part of the spiritual life of the faithful in Gaboltov. The main pilgrimage still takes place on the Saturday and Sunday following the feast of Our Lady of Mount Carmel (July 16). Thousands of pilgrims, many of whom travel on foot, come to Gaboltov from all over Slovakia for this occasion [44].

From Gaboltov, two routes lead to the Basilica of Our Lady of the Seven Sorrows in Šaštín: a northern route of 665 km and a southern route of 804 km. This pilgrimage route was created to increase religious and historical awareness and the visitation of cathedrals, basilicas, and significant places. Both routes of the pilgrimage path form a chain of sites associated with Marian veneration while also introducing pilgrims to many lesser-known natural and cultural attractions of Slovakia [45].

*3.2. Characteristics of Selected Cultural Monuments of Pilgrimage Tourism*

This chapter showcases the most significant and well-known landmarks of pilgrimage tourism in Slovakia from the perspectives of culture, history, and visitation. The significance of the following landmarks is underscored by the fact that they are part of the official route of the aforementioned Way of St. James.

Levoča is known for its tradition of Marian worship, which spans over seven centuries, with the first church mentioned in chronicles from as early as 1247. According to Lányi's chronicle, at that time, the people of Levoča built a small church or chapel on the current site of Olivová Hora. Unfortunately, Lányi's chronicle has not survived. Several historians concur with the opinion that by 1247, a chapel for lepers (leprosarium) may have already stood on Olivová Hora. However, another chronicler of the city, Štefan Meyer, confirms Lányi's assumption in his manuscripts: "... in 1247, the people of Levoča built a small church on Olivová Hora, which they consecrated to the Holy Spirit". Records of canonical visitations also note that "... in 1247, a chapel was built on the hill".

Historian Rudolf Jurík (1948) states, "In 1673, the Spiš provost and the bishop of Nagyvárad (Oradea), Georges Bársony, declared that, according to ancient tradition, the Virgin Mary appeared in glory to a shepherd on Marian Hill. Bársony allegedly verified the legend and can prove its credibility. Nevertheless, no researcher has thus far confirmed this version of the supernatural event on the hill of Levoča" [46].

Hence, there is no immediate mystery associated with Marian Hill (Mariánska Hora). The more probable connection to the origin of the pilgrimage site above Levoča seems to be related to the Tartar invasions, and the beginning of pilgrimages to Marian Hill was historically conditioned by the arrival of the Friars Minor (Franciscan friars) in the town. This religious order began to cultivate and spread the cult of the Virgin Mary. Although Marian Hill lacks a historically researched or proven legend of the Virgin Mary's apparition, it possesses much greater richness. It did not come suddenly in the form of a miraculous gift but developed over centuries. The spiritual dimension and Christian foundation of the sanctuary in Levoča began to take shape with the first pilgrims in the 15th century.

**Basilica of St. James, Levoča**

The ancient pilgrimage tradition in Levoča is represented by the Basilica of St. James, situated in the square (49.026315° S, 20.589190° W), which has been a national cultural monument since 1970. Thanks to the basilica, Levoča was inscribed on the UNESCO World Heritage List in 2009 [14].

This impressive three-nave church was constructed over the course of the 14th century. In 1392, the Chapel of St. George was added to its northern part. At the end of the 15th century, vestibules were added to the northern and southern entrances from the

late 14th century. Among its interior furnishings, alongside two unique late-Gothic altars, special attention should be given to the Via *Dolorum* altar and the older baptistery of the church's furnishings. The wall paintings, with cycles from the 14th to 15th centuries, are remarkable examples of contemporary religious representation. Following a fire in the first half of the 19th century, a new Gothic tower was built at the western part of the church. With its architecture and interior, the church forms a unique collection of historical art and architectural monuments, primarily from the 15th to the 17th century [47].

Inside the church, the magnificent architecture of the main altarpiece stands out. With a height of 18.6 m, it is the tallest in the entire Gothic world and represents the pinnacle of the work of the sculptor **Master Paul of Levoča (around 1465–1537)** [48]. In the predella of the altarpiece is the scene of the **Last Supper**, and the core of the entire altarpiece consists of a shrine containing monumental sculptures of the **Madonna with Child**, **St. James,** and **St. John** in oversized proportions. The altarpiece wings are adorned with reliefs from the life of St. James and St. John, and on their reverse sides, there are panel paintings depicting scenes from the Passion of Christ [49].

While the basilica serves religious purposes daily and attracts tourists and pilgrims with its numerous works of immeasurable artistic and historical value, from the perspective of religious tourism, pilgrims primarily prefer to stay near Marian Hill.

**Basilica of the Visitation of the Blessed Virgin Mary, Marian Hill (Mariánska Hora), Levoča**

The Basilica of the Visitation of the Blessed Virgin Mary on Marian Hill near Levoča (49°02′36″ S, 20°35′52″ W) is one of the oldest pilgrimage sites in eastern Slovakia. A pilgrimage takes place here annually on the first Sunday in July.

The beginnings of the Marian cult in this location date back to the 13th century. In 1247, the first chapel was built as an expression of gratitude to the Blessed Mother for protection against the invasion of the Tatars. The celebration of the Feast of the Visitation of the Blessed Virgin Mary in this area is associated with the arrival of the Franciscans in Levoča in 1311. In 1470, the original chapel was rebuilt into a Gothic church, adorned with a statue of the Virgin Mary in the same style. Pilgrims continued to visit the site even during the Reformation, but the resurgence of pilgrimages occurred during the period of the Counter-Reformation. The first major procession and pilgrimage took place on 2 July 1671. Two years later, with papal permission, the Church of the Visitation of the Blessed Virgin Mary became a place for indulgences. The final reconstruction occurred between 1906 and 1914, resulting in the church's present appearance. The original Gothic statue of the Virgin Mary from the second half of the 15th century still graces the main altar. The new church was consecrated by the Bishop of Spiš, Ján Vojtaššák (1877–1965), on 2 July 1922 [50].

Today, there is also a religious house and accommodation for pilgrims on the site. What was once a pilgrimage site for believers of both Latin and Greek rites from eastern and central Slovakia has gradually become a nationwide pilgrimage site. Even in the past, despite the government's opposition, numerous believers visited during the period of totalitarian rule before 1989.

The significance of this pilgrimage site was recognized by the then-Pope, John Paul II (1920–2005). On 26 January 1984, he bestowed the title of *minor basilica* upon the Church of the Visitation of the Blessed Virgin Mary. On 3 July 1995, the largest pilgrimage to date took place here, with the personal participation of over 650,000 believers. Presently, Marian Hill is visited by approximately two million pilgrims each year [51].

**Basilica of Our Lady of Seven Sorrows, Šaštín**

The Basilica of Our Lady of Seven Sorrows (*minor basilica*), Šaštín (48°38′21″ S, 17°08′35″ W) is one of the most significant basilicas in Slovakia and is simultaneously a national cultural monument. It was built in honor of the Seven Sorrows of the Virgin Mary, whose sculpture is on the main altar of the church.

Construction work began in 1736, based on a design by the renowned architect and priest Máté (Matej) Vépi (1710–1747). By 1756, most of the church's interior was completed, and the decoration of the sanctuary and the nave began, carried out by the distinguished

painter Jean Joseph Chamant (1699–1768). In 1765, the original towers were furnished with bells. Repairs to the main façade were undertaken in 1813, and roof repairs took place in 1851–1852. In 1987, extensive restoration work on the basilica commenced, including the restoration and roofing of both towers, as well as restoration work on paintings and sculptures within the interior [52].

Pilgrimages to the "miraculous" statue of the Seven Sorrows of the Virgin Mary have been taking place since 1564, when Angelika Bakic, the wife of Count Imre Czobor (1520–1581), the owner of the Šaštín estate, had it commissioned. Official pilgrimages began in 1732, when the statue of the Seven Sorrows of the Virgin Mary was made the object of a special cult approved by Archbishop Imre Esterházy (1664–1745) of Esztergom in the same year.

On September 8, 1864, a grand celebration was held in Šaštín, during which the Archbishop of Esztergom, Cardinal János Keresztély Scitovszky (1785–1866), crowned the statue with golden crowns blessed by Pope Pius IX (1792–1878). The event was attended by 300 clergy and over 120,000 pilgrims [53]. Over the past decade, Šaštín has annually welcomed around 20,000 believers during the national pilgrimage, but in the Jubilee Year of 2014 dedicated to the Seven Sorrows of the Virgin Mary, the number of pilgrims reached 50,000. The largest pilgrimages take place on the feast day of Slovakia's patroness, September 15, established in 1966 by Pope Paul VI (1897–1978), who two years earlier elevated the Šaštín church to the status of *minor basilica*. Another significant pilgrimage occurs on the movable feast of Pentecost (Whitsunday). Throughout the regular year, both private and organized pilgrimages take place, offering an intriguing insight into the sociology and structure of pilgrim visits. In addition to the Roman Catholic Church, since 2009, the Greek Catholic Bratislava Eparchy has also been organizing pilgrimages on the Feast of Our Lady of Sorrows, which falls on the last movable feast after Easter. The attendance during individual pilgrimages in Šaštín in selected years is provided in Table 2.

**Table 2.** Pilgrimages to the Basilica of Our Lady of Seven Sorrows, Šaštín (elaborated by authors according to [53]).

| Name of the Pilgrimage | Since | Average Number of Pilgrims | Pilgrimage Date |
|---|---|---|---|
| Pilgrimage of Love | 2007 | 400 | Weekend after St. Valentine's Day |
| Men's Pilgrimage | 2007 | 300–400 | Saturday after Easter Sunday |
| Pilgrimage of Priests | 2008 | 200–300 | Second Saturday in May |
| Mothers' Pilgrimage | 2011 | 100 | Second Sunday in May (Mother's Day) |
| Pilgrimage of Firefighters | 2014 | 250 | Second Saturday in May |
| Pilgrimage of Motorcyclists | 2008 | 300 (counted by motorcycles) | Last Saturday in June |
| Rosary Pilgrimage | 2007 | 1500–3000 | First Saturday in October |

**Description of the selected thematic routes in Šaštín**

**Pilgrimage of Love**

This pilgrimage is dedicated to those who are dating, engaged, or married. At the pilgrimage site, they can give thanks for their relationship, seek advice, and draw from the wisdom of older and more experienced couples. Around the feast of St. Valentine—the patron saint of lovers—a traditional pilgrimage is held for engaged couples, spouses, and young people who wish to live in a pure relationship. During the period around St. Valentine's Day, lasting approximately 2 days, a program is organized for 150 pairs of pilgrimage participants. For registered participants, there are live testimonials of life experiences, interesting lectures, group work with selected couples, musical concerts, theater, and much more. In the Basilica (Šaštín), Holy Masses are celebrated every day during this period, which are also open to the public. Part of this pilgrimage is the making

of vows during Holy Masses, including vows of purity for young (unmarried) participants, where they express their commitment to live in a pure relationship, as well as the renewal of marriage vows. Participating couples can receive advice on how to maintain faith in difficult moments of marriage or partnership, how to educate in faith, how to persevere in faith, and what the roles of husband and wife are in faith [54].

**Pilgrimage of Priests**

In Šaštín, the pilgrimage is annually organized by the Community of Prayer for Priests and is part of a 40-day chain of prayers and fasting priests, which begins on the feast of Our Lady of Sorrows. The full-day pilgrimage program typically commences with introductory remarks and a welcome to guests and pilgrims. The program includes testimonies of priests' life stories, support sessions, and musical performances. One of the poignant moments of the pilgrimage, for example, was in 2018, when the relic of Don Titus Zeman was displayed for public veneration for the first time since his beatification process. The pilgrimage also features the Stations of the Cross, during which testimonies from the lives of those persecuted for their faith and the Church during the time of communism are read. The program is supplemented by morning praises. The culmination of the Pilgrimage for Priests program is the Holy Mass, aimed at encouraging those present to fervently pray for priests who are in need of these prayers—whether due to illness, burnout, doubts, or other issues [55].

**Pilgrimage of Firefighters**

Every year in May, members of the volunteer fire brigade gather in Šaštín, marching through the town. Firefighters come together with their leaders to pay homage to their patron saint and protector, St. Florian, alongside his statue at the National Shrine before the Virgin Mary and her Son and also to deepen their faith. The pilgrimage is attended by ordinary members of volunteer fire brigades from the entire surrounding area together with their leaders—chairpersons of local volunteer fire brigades, directors of district Fire and Rescue brigades, mayors and municipal leaders, as well as representatives and leaders of the Church. On this day, a solemn Holy Mass is celebrated in honor of firefighters, at which attendees express gratitude for their work, dedication, sacrifice, and perseverance in this noble service. This day is a celebration, but also a remembrance and a sacrifice for those who would have wanted to be here but are no longer able, and a great joy and blessing for those who, thanks to the example of firefighters, have found a better life [56].

**Mothers' Pilgrimage**

During the Marian month of May, expectant mothers traditionally come to the National Shrine to the Heavenly Mother to entrust her with their secret and gift from the Lord God. The pilgrimage is not only intended for them. They come here along with their husbands and children, as well as all those who believe that life is a gift from conception onwards. A significant part of the pilgrimage program is the solemn Holy Mass. The theme revolves around the child, family, and life. Following the Holy Mass, the program continues with a lecture and discussion on the topic: "We did not give ourselves life, but we received it". At the end of the program, a small agape is typically prepared [57].

**Pilgrimage of Motorcyclists**

Every June, motorcyclists from across Slovakia participate in the Motorcyclists' Pilgrimage in front of the National Shrine in Šaštín, to collectively seek and express gratitude to the Patroness of Slovakia for protection on the roads. Motorcyclist pilgrims come not only from the surrounding areas but from all over Slovakia. Following the solemn Holy Mass, all motorcycles and their owners are blessed in front of the Basilica, and finally, all participants take a group photograph [58].

**Men's Pilgrimage**

The Saturday after Easter Sunday belongs annually to men of all ages at the Basilica in Šaštín. They come on foot, by bicycle, car, or train, in rain or shine, to pray and express everything that has troubled, pleased, or occupied them throughout the year and to seek help through prayer. A rich program is prepared for these men each year. During this

pilgrimage, lectures and discussions are held where men share their strong life stories and trials, and motivational films are also screened. After these activities, the men kneel in prayer for the Holy Rosary. This is followed by a solemn Holy Mass. The conclusion of the pilgrimage involves signing the Memorial Book at the Basilica and a small social gathering in the monastery's hall. Here, men sit down and chat over refreshments, greet friends they meet here annually, share experiences from the past year, meet new men, and learn of new life stories or experiences [56].

**Rosary Pilgrimage**

The Church celebrates the memorial of the Virgin Mary of the Rosary on Monday, 7 October. On this occasion, a traditional pilgrimage is held in Her honor. The Rosary prayer tradition dates back to the turn of the 12th and 13th centuries. It is said that its author and propagator was St. Dominic, who died in 1221. The Order of Preachers (Dominicans), founded by St. Dominic, furthered his work. The Rosary is one of the most widespread prayers in the Catholic Church worldwide. It consists of four parts, each named after the mysteries it conveys to us: the Joyful Mysteries (events of Jesus' infancy), the Luminous Mysteries (Jesus' public ministry), the Sorrowful Mysteries (Jesus' passion) and the Glorious Mysteries (the Resurrection). Each of these parts comprises five decades. Each decade includes the following prayers: Our Father, Hail Mary, Glory Be, and O My Jesus [56].

**The Church of the Assumption of the Virgin Mary, Calvary Hill, Nitra**

This church, a former monastery, is located at the foot of Calvary Hill (48.298744° S, 18.092013° W). Originally, it was a single-nave Romanesque structure mentioned in a document by Hungarian King Béla IV (1206–1270) from 1248. Later, in the second half of the 18th century, it was rebuilt following the establishment of the Nazarene monastery. The church has a longitudinal floor plan with a transverse nave, a square tower, and two chapels. The four-winged monastery adjoins the southern wall of the church. The monks of the original Spanish Nazarene order came to Nitra in 1766 at the invitation of Nitra Bishop Ján Gustíni-Zubrohlavský (1708–1777) to take care of the church and pilgrims. During this period, the church was a significant pilgrimage site dedicated to the Nitra Mother of God, the wooden Pietà on the side altar of the church. The bishop had a single-story monastery with cells and a courtyard built for the religious community. After the dissolution of the Nazarene order in 1767, they departed. In the second half of the 19th century, the church was renovated in the Neo-Romanesque style, the interior was painted, and the altars were restored. The monastery was also rebuilt. In 1925, members of the Society of the Divine Word (Verbites) settled in the monastery and undertook structural renovations of the church. During this period, an additional floor was added to the ground floor of the monastery, and another wing was built. In 1948, the chapel of St. Theresa, a sacristy, and another wing of the monastery were added to the church. In 2010, the underground spaces of the church were reconstructed, and their original purpose as crypts was restored. The church houses a Pietà sculpture, the "miraculous Mother of God" with the crucified Christ in her arms, dating from the late 17th century and placed on the main altar. It is a 112 cm tall statue, and the first pilgrimages to it began in 1747 [56].

Mass-pilgrimage tourism in this location primarily takes place in the open-air surroundings of Calvary Hill (Kalvária Hill). Every year, thousands of pilgrims from Slovakia and neighboring countries gather here on the Feast of the Assumption of the Virgin Mary (August 15). On the feast of St. Cyril and Methodius (5 July), a national pilgrimage with the participation of Slovak bishops is organized here.

The history of the small sacred architecture of the Stations of the Cross, serving devout pilgrims, dates back to the turn of the 18th and 19th centuries on the hill. After the dissolution of the Kamaldulian monastery on Zobor by Emperor Joseph II (1741–1790) in 1803, the Stations of the Cross reliefs by Austrian sculptor Franz Xaver Seegen (1724–1780) were transferred to the newly built chapels between the Assumption of the Virgin Mary and the hilltop. The original hexagonal chapel that stood on the summit for centuries became the 14th station of the Stations of the Cross, known as the Chapel of the Holy Sepulchre.

Only one of the original Stations of the Cross reliefs has been preserved, which depicts Christ carrying the cross. It is presently located on the northern side of the church above the pilgrim altar. The current chapels were built in 1885 from the church to the summit at regular intervals. The Stations of the Cross reliefs were provided from Munich and financed by Nitra Bishop Augustín Roškoványi (1807–1892). The restored Calvary was consecrated on 8 November 1885 [59].

**Basilica of the Visitation, Staré Hory**

The Basilica of the Visitation is a Roman Catholic pilgrimage church in Staré Hory (48°50′02″ S 19°06′46″ W). The church, which is a national cultural monument, was built between 1448 and 1499. Its main altar features the "graceful statue of the Virgin Mary", to which believers attribute miraculous power. This statue is the reason why Staré Hory has a significant pilgrimage tradition that has lasted for more than half a millennium. What was originally a small mining-settlement parish has become a recognized sacral area where thousands of believers, not only from Slovakia, enter annually. The increasing number of pilgrims, along with a request from the Bishop of Banská Bystrica, František Berchtold (1730–1793), in the past motivated Pope Pius VI to grant Staré Hory special privileges as a pilgrimage site in 1780. The church was declared a *minor basilica* in modern times, on 1 August 1990. During the solemn event, combined with a Mass celebrated by Bishop Rudolf Baláž (1940–2011) of Banská Bystrica and forty priests, approximately 12,000 pilgrims attended [50].

An integral part of this pilgrimage site is Studnička, where local residents hid the statue of the Virgin Mary during the peasant uprisings in the 17th century. In 1711, it was unearthed and carried in a festive procession to the altar. In memory of its concealment, an image of the Virgin Mary was displayed at the site. In the 19th century, the Studnička spring brought healing to the then parish priest, Matej Hrivňák, who, in gratitude, erected a chapel at the site and improved the access road. On 28 June 1942, this pilgrimage site, along with the new statue of the Virgin Mary, was consecrated by Bishop Andrej Škrábik (1882–1950) of Banská Bystrica [50].

*3.3. Attendance at Selected Sites*

**Continuity of Levoča Pilgrimages**

Pilgrimages to the Marian Hill above Levoča have maintained their historical continuity from the mid-13th century to the present day. Believers have come to this sanctuary even during turbulent periods of the Reformation and the moderate Counter-Reformation in Spiš, and they were not discouraged even during the normalization period of the 1970s and 1980s. Even before the COVID-19 pandemic, one of the main Levoča pilgrimages was threatened.

In 1979, everything was prepared for the pilgrimage when an epidemic of viral hepatitis, especially affecting young people, broke out in the districts of Spišská Nová Ves, Stará Ľubovňa, Poprad, Prešov, and Košice. Those affected were immediately hospitalized. Later, it was confirmed that it was Q-fever, an infectious viral hepatitis of type C. As a result, the pilgrimage took place in October on the Feast of the Rosary of the Virgin Mary.

Participation in the pilgrimages to Marian Hill can be quantified in the period from 1945 to 1989 and in the era of free pilgrimages from 1990 to the present. In the first stage from the end of World War II to 1989, **60,000 to 80,000** believers came to Levoča's Hill. This number includes two days of the so-called main pilgrimages, which always occurred on the nearest Saturday and Sunday after 2 July. The state-sponsored power did not allow a large number of Masses on other days. Even in those difficult times, however, the sanctuary of Levoča became known, especially among international circles. The first Slavic Pope, St. John Paul II, marked and emphasized the importance of the Marian Hill with his apostolic brief, which elevated the local church to the status of a *minor basilica* on 26 January 1984 [51]. The increase in attendance at the main Levoča pilgrimages occurred in the 1980s. Even official reports from the then Department for Church Affairs of the District National Committee in Spišská Nová Ves stated that **80,000 to 90,000** individuals attended over two days. According to some unofficial but reliable estimates, this number could have been higher by 20,000 to 30,000 pilgrims [51].

**Participation in Free Pilgrimages**

After November 1989, there was a shift in the attitude toward the Catholic Church and other religions in Slovakia. In the organization of the Levoča pilgrimages, a significant change occurred: daily Masses were added to the weekend following the Feast of the Apostles St. Peter and St. Paul, which falls on 29 June. Believers from all over Slovakia and abroad were thus given the opportunity to visit the Marian sanctuary above Levoča throughout the pilgrimage week from 29 June to the main pilgrimage on Sunday. This significantly increased the number of participants.

According to archival sources from the Parish Office in Levoča and publications by the current emeritus dean and rector of the Levoča basilicas, Monsignor Prof. František Dlugoš, nearly **500,000 people** attended the main pilgrimage on 6 July 1991. During the entire pilgrimage week from the Feast of the Apostles St. Peter and St. Paul on 29 June 1991, to Sunday, 7 July, almost a **million believers** came to Marian Hill. The main weekend pilgrimages to Levoča in the period from 1990 to 1995 were attended by an average of **250,000 to 300,000** people [51]. The first Slavic Pope, St. John Paul II, celebrated a pontifical Mass on 3 July 1995, during his second apostolic visit to Slovakia on the Marian Hill. The media estimated the number of participants immediately after the end of the service to be **600,000** people, but after a subsequent count of distributed hosts and an analysis of photographs, the number increased to nearly **800,000** pilgrims present [51].

The previously limited capabilities of terrain reconnaissance and arrival counting (as civilian helicopter reconnaissance flights were not available, let alone drones) gradually improved. As a result, the statistical and informational outputs from the Levoča pilgrimages became more reliable and verifiable. Before the outbreak of the COVID-19 pandemic, the number of participants during the two main indulgence weekends fluctuated between **250,000 and 300,000 people**, and during the entire pilgrimage week, it reached **half a million participants** [51]. On 4 October 2004, representatives of the largest pilgrimage sites in Europe met in the Hungarian town of Máriapócs. Representatives of the Spiš Diocese personally heard the decision of the *Secrétariat du Réseau Marial Européen*, which recognized Levoča and the Marian Hill as a significant pilgrimage center comparable to Lourdes, Fatima, Częstochowa, and Mariazell.

**Pandemic and the Levoča Pilgrimages**

The global and Slovak crises related to the pandemic and diseases led to the cancellation of the traditional main pilgrimage to Levoča in 2020. The Parish Office in Levoča was allowed to include only one indulgence Mass inside the basilica with strict anti-pandemic measures on the Feast of the Visitation of the Virgin Mary on 2 July. The Mass was broadcast live on Catholic TV Lux. Despite unfavorable rainy weather, **3000 people** participated in the pilgrimage [51].

Due to pandemic measures, the organizers of the pilgrimage to Marian Hill for the year 2021 chose a significantly reduced program over two days, on Friday, 2 July, and Sunday, 4 July, compared with previous years. This was very atypical for the Levoča pilgrimages before the COVID-19 crisis, but the gathering of believers welcomed even this limited form. The almost one-hundred-member team of organizers found valuable understanding, cooperativeness, and discipline among the participants in adhering to anti-epidemic measures. Pilgrims also accepted the necessary exclusion of car and bus transportation to the Levoča sanctuary. The Levoča City Office set up two mobile testing sites at its own expense, where pilgrims could get tested. No participant tested positive out of the total number.

The Parish Office in Levoča included more liturgies in the St. James Basilica in the town during the pilgrimage weekend in 2021, urging believers to use this alternative, with attendance at such Masses also counted as part of the pilgrimage. In total, around **30,000 believers** participated in this combined and still significantly improvised pilgrimage due to the pandemic.

The organizers allowed unlimited access to the Levoča Marian sanctuary with personal cars from Sunday, 26 June, to Friday, 1 July 2022. Approximately one **hundred thousand pilgrims** took advantage of this opportunity during the six indulgence days.

On Saturday and Sunday, participation exceeded **one hundred and fifty thousand partici-pants**, resulting in a total of **four hundred thousand people** in the summary [51].

Surprisingly, many families with young children visited Marian Hill during the weekend. The atmosphere of the pilgrimage was festive, deeply spiritual, prayerful, and friendly that year. No extraordinary events occurred; all liturgies and spiritual events proceeded unusually smoothly and peacefully, magnified by the grandeur of the space and time at the Levoča pilgrimage site, which is part of the association of twenty of the largest European Marian sanctuaries.

The Marian Hill is not only the location of the largest Slovak pilgrimage but also hosts pilgrimages for various groups, including Christian seniors, children, and youth. It is the site of the Marian octave, the pilgrimage of the Christian Democratic Movement, wheelchair users, pilgrims from Zakopane, and seminarians from the Bishop Ján Vojtaššák Priestly Seminary in Spišská Kapitula.

Since 2010, the Levoča Marian sanctuary has been participating in the worldwide Prayer Relay for Priests. The event always takes place during the June feast of the Most Sacred Heart of Jesus. On 7 June 2013, Marian Hill received the Annual Shrine Award from the World Prayer Relay Headquarters for the excellent preparation and conduct of this spiritual event.

Former Bishop of the Spiš Diocese, Monsignor Štefan Sečka, blessed a new Way of the Cross around the basilica on Marian Hill on 29 June 2019. It is a monumental work: each of the fourteen stations of the cross has a weight of 4 tons.

The Levoča Way of the Cross is the result and part of the cooperation between Slovakia and Poland under the Interreg program and the international project "Light from the East", in which four major pilgrimage sites in the Prešov Region participate: Levoča, Gaboltov, Ľutina, and Litmanová. The Levoča Way of the Cross represents a significant contribution to the development of religious tourism in the Prešov Region.

Holy Masses are celebrated every Sunday and on other holidays at Marian Hill from Easter to the end of October of the calendar year. Spiritual exercises for priests and believers are also held at the sanctuary during this season, attracting not only believers but also tourists. The year-round attendance approaches **2 million people** [51].

**Basilica of Our Lady of Seven Sorrows, Šaštín—Participation of the Faithful in Worship and Pilgrimages from 2018–2022**

In 2018, around 165,000 pilgrims visited the basilica (Table 3). Of this total, nearly 57,000 pilgrims participated in state pilgrimages and other events organized by the National Shrine. The most significant pilgrimages with the highest attendance traditionally included the celebrations of Our Lady of Sorrows, the Patroness of Slovakia, with around 40,000 pilgrims, and the Pentecost (Whitsunday) Pilgrimage, with approximately 2800 believers. Additionally, they organized nine other state pilgrimages. The basilica was frequently visited by various groups, including a nationwide pilgrimage of healthcare workers and an archdiocesan pilgrimage of church schools. Parishes and communities also made organized group pilgrimages to the Heavenly Mother. There were 195 such Slovak groups and 32 foreign ones, comprising nearly 12,000 Marian devotees. Around 1800 believers utilized the National Shrine grounds for their spiritual exercises, renewals, and stays [60].

**Table 3.** Visitor attendance trends at the Basilica of Our Lady of Sorrows in Šaštín from 2018 to 2022 (elaborated by authors according to [60–64]).

| Year | Total Attendance | Pilgrimage Attendance | Attendance at Worship Services |
|------|------------------|-----------------------|-------------------------------|
| **2018** | 165,000 | 57,000 | 108,000 |
| **2019** | 170,000 | 59,000 | 111,000 |
| **2020** | 8000 | 1500 | 6500 |
| **2021** | 55,000 | 45,000 | 10,000 |
| **2022** | 32,600 | 15,000 | 17,600 |

In 2019, nearly 170,000 pilgrims visited the basilica. Of this total, nearly 59,000 pilgrims participated in state pilgrimages and other events organized by the National Shrine. The most significant pilgrimages with the highest attendance traditionally included the celebrations of Our Lady of Sorrows, the Patroness of Slovakia, with around 38,000 pilgrims, and the Pentecost (Whitsunday) Pilgrimage, with approximately 2600 believers. Additionally, they organized or co-organized nine traditional state pilgrimages. The basilica was visited by more parish communities and groups, who made organized pilgrimages to the Heavenly Mother in larger or smaller groups. There were 247 Slovak groups and 58 foreign ones, comprising almost 14,000 Marian devotees. Around 1100 believers used the National Shrine grounds for their spiritual exercises, renewals, and stays with a spiritual focus [61].

From March 2020, pandemic containment measures were implemented in Slovakia, which included restrictions on public Masses, later without the presence of the faithful. These measures also affected all planned spiritual activities and events in the basilica, resulting in a lower number of pilgrims that year. Restrictive measures most impacted the National Pilgrimage—the celebrations of Our Lady of Sorrows, the Patroness of Slovakia—which only allowed 1500 pilgrims to participate. Among the traditional state pilgrimages, the Pilgrimage of Love was still carried out according to the normal regime, with the participation of 150 couples. The participation of the faithful on the first Saturdays of the month also alternated depending on the current situation. Pilgrims mostly visited the basilica in small groups, communities, and families. There were 91 such groups registered, with a total of around 8000 pilgrims throughout the year. On the other hand, there was an increase in the number of individual pilgrims who visited on their own, either to see the basilica, pray silently, or to avail themselves of the spiritual ministry of the priests [62].

The year 2021 was marked by an unfavorable pandemic situation in Slovakia. Except for the summer, there were virtually continuous restrictions regarding public Masses. These restrictions also affected all planned spiritual activities and events in the basilica. Thanks to internet broadcasting through the Studio 7bolestná (Seven Sorrows) channel, Holy Masses and other spiritual programs were broadcast live on YouTube. Believers had the opportunity to participate in the celebration of Holy Masses through this online platform. Thanks to the generosity of donors and supporters, the studio was expanded, and the quality of the broadcasts improved. In the fall of 2021, an extraordinary event of national significance took place when on 15 September, the Feast of Our Lady of Sorrows, the Patroness of Slovakia, the faithful had the rare opportunity to welcome Pope Francis. Approximately 45,000 pilgrims from all over Slovakia attended the pontifical Mass, albeit with restrictions. Among the traditional pilgrimages, only the Mothers' Pilgrimage, the Motorcyclists' Pilgrimage, the Greek Catholic Pilgrimage of the Bratislava Eparchy, the Rosary Pilgrimage, the Priest's Pilgrimage, and the Pentecost (Whitsunday) Pilgrimage were realized. Believers could attend these Holy Masses in person, though with restrictions [63].

Pandemic containment measures were in place in Slovakia until February 2022, which limited the participation of the faithful in worship services. Believers who could not attend the basilica due to these restrictions could watch the Holy Masses and other spiritual programs through live internet broadcasts. The most significant pilgrimages with the highest attendance included the traditional celebrations of Our Lady of Sorrows, Patroness of Slovakia, on September 15th, with around 15,000 believers in attendance. The Pentecost (Whitsunday) Pilgrimage had approximately 1200 participants. Throughout the year, during the first Saturdays of the month (Fatima Saturdays), about 3100 believers made pilgrimages to the basilica. Apart from the Pilgrimage of Love, the other state pilgrimages were already being organized without limitations, and around 1000 pilgrims participated in them. The basilica was also visited by parishes and communities that came on foot, by bicycle, by car, and by bus, in both larger and smaller organized groups. There were 121 such Slovak groups and 24 foreign ones, comprising approximately 8500 people who made the pilgrimage [64].

## 4. Discussion

The pilgrimage of Levoča has preserved its historical continuity from the mid-13th century to the present day. Believers have come to this sanctuary even through the turbulent periods of the Reformation and Counter-Reformation (relatively moderate in the Spiš region), and they were not deterred even by the normalization process in the 1970s and 1980s during the 20th century.

In the initial phase, spanning from the conclusion of World War II to 1989, Levoča Mountain witnessed the pilgrimage of 60,000 to 80,000 believers. The elevation of the local church to the status of a minor basilica by the first Slavic pope, St. John Paul II, underscored the significance of Marian Hill. This pivotal act played a crucial role in the surge of attendance from the primary Levoča pilgrimages during the 1980s. The dissolution of the Socialist Bloc in 1989 triggered a "reorientation" of the Catholic and other churches in Slovakia [51]. This period witnessed a noteworthy transformation in the organization of the Levoča pilgrimages; in addition to weekends, daily masses were introduced, commencing on the feast of the apostles St. Peter and St. Paul on 29 June. This adjustment afforded believers from across Slovakia and beyond the opportunity to partake in the pilgrimage week, extending from 29 June to the main Sunday pilgrimage. The tangible impact of this expansion was reflected in the substantial increase in participants. According to archival records from the Parish Office in Levoča, the main pilgrimage in 1991 saw the participation of nearly 500,000 individuals. This attests to the fact that the introduction of pilgrimage tourism options succeeded in augmenting attendance by nearly tenfold.

Pope St. John Paul II presided over a pontifical Holy Mass on Marian Hill during his second apostolic journey to Slovakia in 1995. Initial media estimates pegged the participants at 600,000 immediately after the service. However, after a subsequent recount of distributed hosts and a meticulous analysis of photographs, the number was revised upwards to almost 800,000 pilgrims present. This serves as clear evidence that the presence of a prominent personality once again led to a substantial increase in attendance, by almost 1.5 times. The Levoča pilgrimages boast a lengthy and enriched history, persisting since their initiation in the 13th century, resiliently continuing even through turbulent periods in Slovakia's historical narrative.

There are several reasons why the Levoča pilgrimages have endured to this day. Among these reasons are the following:

(a.) The origin of the Levoča pilgrimages. The origin of the Levoča pilgrimages is associated with alleged Marian apparitions to two shepherds.

(b.) The spiritual atmosphere of Marian Hill, Levoča. The environment of Marian Hill contributes to its spiritual atmosphere, attracting pilgrims from all over Slovakia and abroad.

(c.) The high religiosity of the population. The high religiosity of the population is reflected in increased attendance at religious events, including the Levoča pilgrimages.

(d.) Papal visits in 1984 and 1995. Papal visits in 1984 and 1995, when Pope John Paul II visited the Levoča sanctuary, significantly contributed to the increased popularity of the shrine, attracting pilgrims from around the world.

Efforts to cultivate the Levoča pilgrimages necessitate strategic actions such as enhancing the infrastructure at Marian Hill, diversifying spiritual and cultural offerings for pilgrims, and fostering stronger collaborations between the church and state authorities in pilgrimage organization. A paramount consideration in this endeavor is the need to enhance the infrastructure at Marian Hill, a critical step to accommodate a substantial influx of pilgrims. Key priorities encompass the establishment of new parking facilities, the enhancement of access roads, and the refurbishment of accommodation facilities. Quality management, which includes a quality team of people with international experience, is also very important [65]. Managers and employees who have the ability to adapt quickly and have intercultural competences dispose with the need for a competitive advantage [66].

In recent years, the attendance at the basilica in Šaštín has been stable. The majority of visitors come for pilgrimages held throughout the year, with notable events being the celebrations of Our Lady of Sorrows, the Patroness of Slovakia, and the Pentecost

(Whitsunday) Pilgrimage. In 2020, there was a significant decrease in attendance due to the coronavirus pandemic. Pilgrimage sites were only open to individual visitors, and organized pilgrimages were prohibited. Overall attendance dropped by only a few percent compared to previous years.

In 2021, the situation with the pandemic improved, and organized pilgrimages were allowed. The basilica was open to groups of visitors. Overall attendance increased to 15,000 visitors. The visit of Pope Francis, who participated in the celebration of Our Lady of Sorrows in Šaštín, attracted the most visitors.

In 2022, the basilica's attendance experienced a renewed increase, reaching 32,600 visitors. All pilgrimages were permitted, including the Pilgrimage of Love, which had been canceled in 2020 and 2021. However, the return of visitors after the pandemic period was not as swift as anticipated. There were higher expectations due to the assumption that individuals during the COVID-19 pandemic would exhibit a heightened inclination toward spirituality and spiritual life. Factors such as people's cautious approach and the resurgence of the pandemic might have influenced this slower return. The results of this study do not primarily address measures taken during and immediately after the pandemic, as several studies have already been published on this topic [67–70]. We focused on the return of visitors to pilgrimages and the factors influencing them. It can be assumed that even another pandemic will not disrupt the functioning of pilgrimage tourism from a long-term perspective and that the number of visitors will remain stable. A significant increase in pilgrimage visitors can be expected primarily through visits by prominent figures, mostly from the ecclesiastical sphere, such as the Pope's visit to Slovakia. These findings have practical implications for pilgrimage organizers, who should focus on effective marketing. However, it is also important to consider that pilgrims are there for spiritual experiences. Overexposed tourism products and services may add recreational value, but on the other hand, they may be disruptive [71,72].

## 5. Conclusions

Pilgrimages represent a historically and spiritually significant phenomenon that has persisted through the ages. Their enduring nature is underpinned by various factors, encompassing historical importance, religious significance, and the creation of a pleasant atmosphere. The pilgrimage site at Marian Hill holds the potential to evolve into one of the pre-eminent pilgrimage destinations in Central Europe. The development of pilgrimage tourism to Marian Hill stands poised to contribute significantly to the broader tourism landscape in the region.

Strengthening collaboration between ecclesiastical and governmental bodies is instrumental to ensuring the seamless execution of pilgrimages and to garnering increased support from the state. Pilgrimage attendance has been notably influenced by the ongoing challenges posed by the coronavirus pandemic. Over the past decade, attendance has generally remained stable. However, 2020 witnessed a pronounced decline due to pandemic-related restrictions, with the basilica limited to individual visitors and organized pilgrimages suspended. In 2021, with improvements in the pandemic situation, organized pilgrimages were reinstated, resulting in a marked surge in attendance, albeit taking nearly three years to revert to previous levels.

Pilgrimages constitute an integral facet of Slovakia's spiritual milieu, drawing significant popularity and attracting a multitude of pilgrims annually. There is a strong likelihood that this trend will persist in the foreseeable future. The most notable surge in attendance is attributable to visits by prominent religious figures, as exemplified by the Pope. Conversely, the principal threat to the continuity of pilgrimages arises from the implementation of safety measures during pandemic periods.

At the beginning of the research, it was assumed that people, during challenging life periods such as the COVID-19 pandemic, turn to spirituality and religion. Therefore, we anticipated that after the restrictions were lifted, the attendance of pilgrimages would be higher

than before the COVID pandemic, which was not confirmed. On the other hand, our findings revealed that pilgrimage attendance reverted to pre-pandemic levels within three years.

Given that the article investigates the progression of pilgrimage-tourism attendance pre- and post-pandemic, as well as during the pandemic, with a specific focus on two illustrative cases in Slovakia (Levoča and Šaštín), forthcoming research could delve into additional pilgrimage destinations within Slovakia or conduct a comparative analysis of pilgrimage tourism development across diverse European nations.

**Author Contributions:** Conceptualization, M.M. and E.K.; validation, M.J.; formal analysis, D.T.; investigation, E.K.; resources, D.T.; writing—original draft preparation, M.J. and E.K.; writing—review and editing, M.M. All authors have read and agreed to the published version of the manuscript.

**Funding:** This research received no external funding.

**Data Availability Statement:** Data are contained within the article.

**Acknowledgments:** The authors would like to thank Jozef Lapšanský for his exceptional cooperation and constant support, for the time he devoted to us, and for the guidelines, comments, and professional advice that moved us in the right direction when creating this article. This article was created with the support of the project "Environmental Specifics of Selected Mining Water Management Systems in Slovakia" (VEGA 1/0667/21).

**Conflicts of Interest:** The authors declare no conflicts of interest. The funders had no role in the design of the study; in the collection, analyses, or interpretation of data; in the writing of the manuscript; or in the decision to publish the results.

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
