# Peer review of "Development of Pilgrimage Tourism in Slovakia over the Past Decades: Examples of Selected Pilgrimage Sites"

_heritage, doi:10.3390/heritage7030085_

Round 1
Reviewer 1 Report
Comments and Suggestions for Authors
1. better avoid phrases such as "both in our country and worldwide", you may write "both in Slovakia and worldwide"
2. there is a theoretical problem in "Religious tourism, also known as pilgrimage, sacred, faith or spiritual tourism, is a 39 widespread form of tourism" .
it seems that the terms "religious tourism" and "pilgrimage" are identical |(see also line 55). There is no notion of sub-categories, mixed forms within the umbrella term "religious tourism" etc; there is a lack of theoretical analysis of the terms-concepts in the international literature. This part (lines 1-63) should be re-elaborated or deleted and start from line 64.
3. very problematic phrase both sociologically and theologically: "Religious tourism can serve as an excel- 141 lent tool for understanding faith. For believers, church and sacred places are oases for 142 relieving stress and finding new mental well-being"
4. the section 2. Significant Pilgrimage Sites Around the World is very superficial and serves not the article. What about Jerusalem as the ancient pilgrimage destination of all Christians? The pilgrimages of the Orthodox Christians? The Santiago de Compostela? Do the Protestands accept the term "Pilgrimage" for Wittenberg?
5. what does it mean? "the concept of 296 religious tourism is primarily used in the Christian world and has been accepted by the 297 Roman Catholic Church" " Is it pilgrimage meant here? and also "and has been accepted by the 297 Roman Catholic Church". What is the meaning of this phrase?
6. "In the first 601 stage from the end of World War II to 1989, 60,000 to 80,000 believers came to Levoča's 602 Hill. " Where are the data from?? please provide a reference. (see also lines 781-782)
7. The whole 4. Discussion section has a press like character and it needs a deeper analysis. There is the phrase "However, the return of visitors post the pandemic period 840 was not as swift as anticipated. There were higher expectations, assuming that individu- 841 als, during the COVID-19 pandemic, would exhibit a heightened inclination towards spir- 842 ituality and spiritual life. Factors such as people's cautious approach and the resurgence 843 of the pandemic might have influenced this slower return.". There are plenty of articles analysing the post-covid trends in religious tourism, the implications of the measures, the RT resilience etc.
8. The dialogue with existing research should be better and more developed, and more specifically on the impact of the couvid epidemic on the development of religious tourism. It is also suggested that the authors should limit long historical reports that resemble "tourist presentation" and insist more on the factors that enhance the development of religious tourism.
Author Response
Review 1
Cover letter
Dear reviewer,
Thank you for your time and for any comments that help improve our contribution. We hope that the adjustments we have made are in line with your suggestions and comments.
Comments:
- better avoid phrases such as "both in our countryand worldwide", you may write "both in Slovakia
Thank you for your suggestion. It has been revised in accordance with your comment (line 36-37).
- there is a theoretical problem in "Religious tourism, also known as pilgrimage, sacred, faith or spiritual tourism, is a 39 widespread form of tourism" .
it seems that the terms "religious tourism" and "pilgrimage" are identical |(see also line 55). There is no notion of sub-categories, mixed forms within the umbrella term "religious tourism" etc; there is a lack of theoretical analysis of the terms-concepts in the international literature. This part (lines 1-63) should be re-elaborated or deleted and start from line 64.
The definitions of the terms "religious tourism" and "pilgrimage," among others, have been expanded. This section (lines 1-63) has been revised.
- very problematic phrase both sociologically and theologically: "Religious tourism can serve as an excel- 141 lent tool for understanding faith. For believers, church and sacred places are oases for 142 relieving stress and finding new mental well-being"
Problematic phrase: “Religious tourism can serve as an excellent tool for understanding faith. For believers, churches and sacred places are oases to relieve stress and find new mental well-being" has been removed and supplemented with new text.
- the section 2. Significant Pilgrimage Sites Around the World is very superficial and serves not the article. What about Jerusalem as the ancient pilgrimage destination of all Christians? The pilgrimages of the Orthodox Christians? The Santiago de Compostela? Do the Protestands accept the term "Pilgrimage" for Wittenberg?
The section “2. Important pilgrimage sites in the world“ has been removed.
- what does it mean? "the concept of 296 religious tourismis primarily used in the Christian world and has been accepted by the 297 Roman Catholic Church" " Is it pilgrimage meant here? and also "and has been acceptedby the 297 Roman Catholic Church". What is the meaning of this phrase?
The interpretation of this phrase was unclear, leading to its removal from the text.
- "In the first 601 stage from the end of World War II to 1989, 60,000 to 80,000 believers came to Levoča's 602 Hill. " Where are the data from?? please provide a reference. (see also lines 781-782)
The data are from Mr. M. Lapšanský, (personal communication, May 3, 2023), this source has been added to the references (Reference no. 51, also in Acknowledgments).
- The whole 4. Discussion section has a press like character and it needs a deeper analysis. There is the phrase "However, the return of visitors post the pandemic period 840 was not as swift as anticipated. There were higher expectations, assuming that individu- 841 als, during the COVID-19 pandemic, would exhibit a heightened inclination towards spir- 842 ituality and spiritual life. Factors such as people's cautious approach and the resurgence 843 of the pandemic might have influenced this slower return.". There are plenty of articles analysing the post-covid trends in religious tourism, the implications of the measures, the RT resilience etc.
The discussion has been enhanced with a review of post-COVID trends from additional references. However, it's important to note that this article primarily emphasizes the recurrent visits of pilgrims and the factors that impact them.
- The dialogue with existing research should be better and more developed, and more specifically on the impact of the couvid epidemic on the development of religious tourism. It is also suggested that the authors should limit long historical reports that resemble "tourist presentation" and insist more on the factors that enhance the development of religious tourism.
The section 4 Discussion has been supplemented by dialogue with existing research. The text has been modified according to the suggestions and comments (lines 802 - 811).
Reviewer 2 Report
Comments and Suggestions for Authors
After reading the manuscript and paying attention to the data and methods, I rated the manuscript on attributes using the MDPI scorecard. The eventual recommendation of mine is to accept the paper after major revision.
This article gives insight in the religious tourism in Slovakia. The authors have clearly stated the geographical boundaries of the considered topic but unfortunately the aim of study and the research questions are not formulated. The INTRODUCTION should state the purpose of a study and explain the research question.
The title of this article is comprehensive.
The ABSTRACT contains information to enable the reader to understand what was done. My suggestion is to demonstrate clearly the importance of the article to the field.
The KEYWORDS accurately reflect the content.
The INTRODUCTION clearly establishes the context of the study. See line 133: …states that a part of cultural…. I suppose the bold used is a misprint. This section does not provide critical and constructive analysis of existing published literature in a field. The authors should demonstrate the novelty of their research to the field.
The text of SIGNIFICANT PILGRIMAGE SITES AROUND THE WORLD should be deleted. It is too general, just a list of world-wide know pilgrimage sites. It tells nothing. The authors should give explicit description of methods of the research and of how data were collected and analysed. In other words, the section of METHODOLOGY should be included in the article.
In the section of ANALYSIS OF PILGRIMAGE IN SLOVAKIA, the authors clarify the fundamental concepts used in legislation regarding cultural heritage in Slovakia. In this article, the authors interpret pilgrimage tourism in a broader sense, encompassing not only travel motivated exclusively by religious reasons but also the exploration of sacred sites that are simultaneously cultural heritage sites.
The text of PILGRIMAGES AND PILGRIMAGE ROUTES IN SLOVAKIA is clear and easy to read.
In the section of CHARACTERISTICS OF SELECTED CULTURAL MONUMENTS OF PILGRIMAGE TOURISM, the authors give general information on some pilgrimage sites in Slovakia. The criteria of selection of these sites should be explained clearly already in the first paragraph of this section.
I found particularly interesting the section ATTENDANCE IN SELECTED SITES.
The section of DISCUSSION just describes the findings, but I would like to remind that it should analyse and interpret the findings. The authors need to interpret and evaluate what their findings mean, particularly in relation to the research question. They should discuss whether these findings agree with current research? This section should demonstrate the significance of the findings as well. The authors should explain new understanding or insights that emerged as a result of their research.
In the CONCLUSION, the main argument of the paper should be more accurately restated. The conclusions should be consistent with the arguments presented.
Author Response
Review 2
Cover letter
Dear reviewer,
Thank you for your time and for any comments that help improve our contribution. We hope that the adjustments we have made are in line with your suggestions and comments.
Comments:
The INTRODUCTION should state the purpose of a study and explain the research question.
A new chapter (2. Materials and Methods) has been created where the purpose of the study and the research question has been explained.
The INTRODUCTION clearly establishes the context of the study. See line 133: …states that a part of cultural…. I suppose the bold used is a misprint. This section does not provide critical and constructive analysis of existing published literature in a field. The authors should demonstrate the novelty of their research to the field.
The incorrect bold font has been rectified, and this section has been enhanced with additional literature.
The text of SIGNIFICANT PILGRIMAGE SITES AROUND THE WORLD should be deleted. It is too general, just a list of world-wide know pilgrimage sites. It tells nothing. The authors should give explicit description of methods of the research and of how data were collected and analysed. In other words, the section of METHODOLOGY should be included in the article.
The text regarding "SIGNIFICANT PILGRIMAGE SITES AROUND THE WORLD" has been removed. Additionally, Section 2, "Materials and Methods," has been elaborated and incorporated into the article.
The section of DISCUSSION just describes the findings, but I would like to remind that it should analyse and interpret the findings. The authors need to interpret and evaluate what their findings mean, particularly in relation to the research question. They should discuss whether these findings agree with current research? This section should demonstrate the significance of the findings as well. The authors should explain new understanding or insights that emerged as a result of their research.
The discussion has been enhanced with a review of post-COVID trends in references. However, this article primarily centres on the recurring visits of pilgrims and the factors influencing them. Section 4, the Discussion, has been enriched through dialogue with existing research, and the text has been adjusted based on feedback.
In the CONCLUSION, the main argument of the paper should be more accurately restated. The conclusions should be consistent with the arguments presented.
The conclusion has been expanded (lines 802 - 811), reiterating the main argument of the paper.
Reviewer 3 Report
Comments and Suggestions for Authors
The article deals with the development of pilgrimage tourism in Slovakia in the last two decades on the basis of selected pilgrimage sites. In my opinion, it is a valuable study.
Doubtful is part 2. In this part several religious tourism sites of different religions are described. The elimination of this part should be considered. The overview of the issue of religious tourism seems sufficient.
There were technical errors in many places: incorrect spaces.
Author Response
Review 3
Cover letter
Dear reviewer,
We appreciate your time and valuable feedback that have contributed to enhancing our work. We have carefully considered your suggestions and comments, and we believe that the adjustments we have made align with your recommendations.
Comments:
Doubtful is part 2. In this part several religious tourism sites of different religions are described. The elimination of this part should be considered. The overview of the issue of religious tourism seems sufficient.
The section “2. Important pilgrimage sites in the world“ has been removed.
There were technical errors in many places: incorrect spaces.
Technical errors have been corrected throughout the entire manuscript.
Reviewer 4 Report
Comments and Suggestions for Authors
I have read the the article “Development of Pilgrimage Tourism...” with great interest as it is a desciption of moderrn pilgrimage in an – to me - unknown geogrphical field.
However; the greatest problem with the article is the lack of a distinct problem formulation. The article discusses the relation between pilgrimage and tourism, the relationn between heritage (UNESCO) and pilgrimage sites, the effects of the pandemic on the pilgrimage in Slovakia, the general development of pilgrimage in modern Slovakia without connecting these fields and problems to each other. It presents a lot of mumbers connected to this. All of these questions of importance within research on modern pilgrimage, ut need to be related
Perhaps because of the lack of problem formulatiion these various issues discussed do not relate, (and also seems to have been written individually by the 4 authors.) I do think that a discussion of common problem formulation would be helpful in relating the various questions and problems
The overveiw of “Significant Pilgrimage Sites Around the world seems redundant and brings nothing new and seems irrelevant to the problems discussed in the article.
I have two points that in my opinion would bring some of the discussions about pilgrimage in the modern world more up to date. Table 2 presents an overview of the number of pilgrims to the Basilica of Our Lady ... The various groups presented in this table really awakens my curiosity; What is, and who participates in the Pilgrimage of Love, and what is Mothers’ Pilgrimage, and why have the Firefighters its own pilgrimage and what is the point of the Motorcyclists’ pilgrimage and what is Rosary Pilgrimage? I do think a discussion and description of this great variation within pilgrimage in modern Slovakia would be of great interest. I also miss a discussion relating the (Increasing?) interest in pilgrimage related to the political and national development in the nation of Slovakia.
My second idea is perhaps besides the authors’ interest – but came to my mind when reading this article, and would perhpas bring some dynamic to the discussion. In my opinion holy sites are not holy in themselves, the establishment and maintenance of pilgrimage and holy sites demand a lot of work, (by locals, priests, tourism agents etc) the sites are not just sacred and remain so, it craves a lot of work to let them remain holy sites in the minds of people. I recommend the book Pilgrimage in the market place by the British scholar of religion Ian Reader (Routledge 2014) in this connection. What do locals and chuch people do to maintain the sacredness of these places?
Author Response
Review 4
Cover letter
Dear reviewer,
Thank you for your time and for any comments that help improve our contribution. We hope that the adjustments we have made are in line with your suggestions and comments.
Comments:
However; the greatest problem with the article is the lack of a distinct problem formulation. The article discusses the relation between pilgrimage and tourism, the relationn between heritage (UNESCO) and pilgrimage sites, the effects of the pandemic on the pilgrimage in Slovakia, the general development of pilgrimage in modern Slovakia without connecting these fields and problems to each other. It presents a lot of numbers connected to this. All of these questions of importance within research on modern pilgrimage, ut need to be related
The article has been enhanced with a precise problem formulation in Chapter 2, "Materials and Methods." Furthermore, the article has been restructured to ensure cohesion and coherence among its individual chapters
Perhaps because of the lack of problem formulatiion these various issues discussed do not relate, (and also seems to have been written individually by the 4 authors.) I do think that a discussion of common problem formulation would be helpful in relating the various questions and problems
The issues have been delineated in Chapter 2, while in the discussion section, we endeavored to establish connections between the findings and highlight any novel insights derived from the study.
The overveiw of “Significant Pilgrimage Sites Around the world seems redundant and brings nothing new and seems irrelevant to the problems discussed in the article.
The section “2. Important pilgrimage sites in the world“ has been removed.
I have two points that in my opinion would bring some of the discussions about pilgrimage in the modern world more up to date. Table 2 presents an overview of the number of pilgrims to the Basilica of Our Lady ... The various groups presented in this table really awakens my curiosity; What is, and who participates in the Pilgrimage of Love, and what is Mothers’ Pilgrimage, and why have the Firefighters its own pilgrimage and what is the point of the Motorcyclists’ pilgrimage and what is Rosary Pilgrimage? I do think a discussion and description of this great variation within pilgrimage in modern Slovakia would be of great interest. I also miss a discussion relating the (Increasing?) interest in pilgrimage related to the political and national development in the nation of Slovakia.
The overview of different pilgrim groups at the Basilica of the Virgin Mary, as presented in Table 2, has been augmented in Chapter 3.2, "Characteristics of Selected Cultural Monuments in Pilgrimage Tourism." (lines 446-524)
The aim of this article was to identify factors influencing the number of pilgrims in Slovakia in recent years. Your comment regarding the interest in pilgrimage in relation to the political and national development of the Slovak nation is very interesting. However, pilgrimage attendance did not show any change (increase) in connection with political and national developments. This may be due to the fact that significant political or national changes have not occurred in recent decades. The only significant change occurred at the turn of the 1980s and 1990s, when there was a change in the political regime and the establishment of an independent republic in a short period. However, these changes are described in the article (lines 740-757).
My second idea is perhaps besides the authors’ interest – but came to my mind when reading this article, and would perhpas bring some dynamic to the discussion. In my opinion holy sites are not holy in themselves, the establishment and maintenance of pilgrimage and holy sites demand a lot of work, (by locals, priests, tourism agents etc) the sites are not just sacred and remain so, it craves a lot of work to let them remain holy sites in the minds of people. I recommend the book Pilgrimage in the market place by the British scholar of religion Ian Reader (Routledge 2014) in this connection. What do locals and chuch people do to maintain the sacredness of these places?
Your second idea is indeed intriguing, and we agree with it. We believe that delving deeper into the intricacies of pilgrimage management, including organization and community involvement, warrants further investigation and could serve as a topic for a separate article. In this study, our aim was to highlight the pertinent factors and emerging trends in travel tourism that necessitate consideration for future destination management. Therefore, we only briefly touched upon tourist management in this article (lines 777-784).
We appreciate your recommendation of the book "Pilgrimage in the Market Place" by the esteemed British scholar of religion, Ian Reader (Routledge, 2014). It has been incorporated into the text and included in the references (reference no. 67)
Round 2
Reviewer 1 Report
Comments and Suggestions for Authors
no further suggestions
Reviewer 2 Report
Comments and Suggestions for Authors
All comments are addressed. A decision regarding the revised manuscript: accept.
Reviewer 4 Report
Comments and Suggestions for Authors
My objetions and redcommendations had been taken care of and that the article may be printed.